# Enhancing household soybean processing and utilization in the Eastern Province of Zambia, a concurrent triangulation study design

Priscilla Funduluka[1]*, Twambo Hachibamba[2], Mercy Mukuma[2], Phoebe Bwembya[3], Regina Keith[4], Chiza Kumwenda[2], Natasha Muchemwa Mwila[5,6]

1 Department of Public Health, Public Health Nutrition Unit, School of Public Health and Environmental Sciences, Levy Mwanawasa Medical University, Lusaka, Zambia, 2 Department of Food Science and Nutrition, School of Agriculture, University of Zambia, Lusaka, Zambia, 3 Department of Community and Family Medicine, School of Public Health, University of Zambia, Lusaka, Zambia, 4 School of Life Sciences, University of Westminster, London, United Kingdom, 5 Department of Plant Science, University of Zambia, Lusaka, Zambia, 6 German Development Cooperation, Agriculture and Food Security Cluster, Lusaka, Zambia

* pfunduluka04@gmail.com

**Data Availability Statement:** All relevant data are within the manuscript and its Supporting information files.

## Abstract

The purpose of this study was to establish best ways of improving household soybean processing and utilization in selected districts in the Eastern Province of Zambia. This was a concurrent triangulation study design, nested with a cross sectional survey and barrier analysis. Up to 1,237 households and 42 key informants participated in the quantitative and qualitative studies respectively. Quantitative data was analysed using Stata MP 15 software (StataCorp, College Station, TX, USA). NVIVO QSR10 software (QSRInt, Melbourne Australia) was used to organize qualitative data which was later analysed thematically. In this study whole soybean processing and utilization in eastern province was at 48%. However, accessibility to soybean for household consumption throughout the year was negligible (0.29%). Based on the food systems an interplay of factors influenced soybean processing and utilization. In the food environment, a ready-made Textured Soya Protein mainly imported [1,030/1237(83%)] and a milled whole soybean-maize blend AOR 816.37; 95%CI 110.83 to 6013.31 were preferred. Reports of labour intensity, hard to cook properties, coarse milling and beany flavour with associated anti-nutrients negatively influenced whole soybean utilization. In the enabling environment, soybean production AOR 4.47; 95%CI 2.82 to 7.08 increased the chances of utilization. Lack of inputs, poor access to affordable credit and lack of ingredients were deleterious to utilization. Low coverage of existing projects and poor access to technologies were other adverse factors. Among the Socioeconomic factors, a higher social hierarchy shown by owning a bed AOR 1.75; 95%CI 1.22 to 2.49, belonging to the Chewa community AOR 1.16; 95%CI 1.08 to 0 1.25, gender of household head particularly male AOR 1.94; 95%CI 1.21 to 3.13, off farm income and livestock ownership were supportive to soybean utilization. Unfavourable factors were; belonging to any of the districts under study AOR 0.76; 95%CI 0.58 to 0.98, lack of knowledge (55.65%),

**Funding:** This work was made possible with the financial and substantive support of the GIZ in Lusaka, Zambia. The GIZ were also involved in the Conceptualization, Visualization and shaping the methodology.

**Competing interests:** I hereby declare, on behalf of all authors, that there are no financial, personal, or professional interests that could be construed to have influenced the work. This does not alter our adherence to PLOS ONE policies on sharing data and materials.

low involvement of the male folks AOR 0.47; 95%CI 0.30 to 0.73 and belonging to a female headed household AOR 1.94; 95%CI 1.21 to 3.13. Age, time and household size constraints as well as unreliable soybean output markets, lack of land, poor soils in some wards and poor soybean value chain governance were other negative factors. Immediately in the food environment there is need to boost milling of whole soybean while strengthening cooking demonstrations, correct processing, incorporation of soybean in the local dishes and conducting acceptability tests. In the enabling environment, there should be access to inputs, affordable credit facilities and subsidized mineral fertilisers. Post-harvest storage, collective action with full scale community involvement and ownership should be heightened. Socio-economic approaches should target promotion of soybean processing and utilization among all ethnic groups, participation of male folks and female headed households as well as advocating for increased nutrition sensitive social protection. In the medium or long term, capacity building, market development, import substitution agreements, creation of new products, development of cottage industries, information exchange and inter district trade as well as more public-private partnerships and more local private sector players should be bolstered. Lastly farm diversification should be supported.

## Introduction

Soybeans have become a staple part of the human diet because they are nutritionally excellent and contain various functional components that provide a health benefit beyond basic nutrition [1, 2]. Botanically known as Glycine max, soybean belongs to the leguminous family because of its ability to form nodules and fix nitrogen in the soil [3]. Soybean is also a climate-resilient, low-cost crop with food security potential [4, 5]. The crop is predominantly grown in the Western Hemisphere (80–85%) [6]. In Sub-Saharan Africa, South Africa was reported as the largest soybean producer in 2016, followed by Nigeria, Zambia, and Uganda [7]. Zambia is the second-largest soybean producer in Southern Africa, with the Eastern province though comprised of small scale farmers, being one of the three main provinces involved in soybean production [8–10]. Soybean is the most nutritionally rich crop as its dry seed contains the highest protein and oil content among grain legumes [11]. In terms of contribution to dietary intakes of children and adults, it can significantly complement attainment of average daily level of nutrient intakes also known as Recommended Dietary Allowances (RDAs) [12–14]. This is so because it contains considerable quantities of carbohydrates (32%), proteins (40%) and lipids (22%) [15]. Soybeans are also a good source of several vitamins and minerals, including vitamin K1, folate, thiamine, copper, manganese and phosphorus [15–17]. Carbohydrates, protein and lipids in soybean contribute energy amounting to 475 kilocalories with fat contributing the largest proportion followed by proteins and carbohydrates [12, 17]. The energy could add to the daily energy pull of individuals particularly children aged 1–10 years whose energy needs increase with advance in years as well as males aged 15 to 18 years whose energy needs are more. Soybean protein provides several therapeutic benefits as the crop contains most of the essential amino acids in the amounts needed for health [18]. Essential amino acids in soybean include; histidine, isoleucine, leucine, lysine, methionine, phenylalanine, threonine, tryptophan and valine [13, 19]. Soybean protein has potential to contribute towards meeting daily requirements of individuals particularly children aged 7–12 months followed by those aged 1–13 years with high protein requirements, as well as the physically active adolescents and adults [13]. Soy protein however is limiting in Sulphur containing amino acids

methionine and cysteine. This would limit humans to make use of other abundant essential and non-essential amino acids thereby limiting growth and milk production in lactation [20, 21]. Soybean products therefore should be combined with other plant proteins such as those contained in whole cereals like maize to have a complete protein [19, 22]. As a good source of several vitamins and minerals, soybean could also contribute towards the age specific daily needs of minerals and Vitamins if well utilized [16, 17, 23].

Soybean has unique characteristics as it can be made into a variety of products for income generation as well as for household food security [24]. World-wide up to 85% of soybean produced is channelled into animal feed with the remaining being used for human consumption [25]. Popular household and industrial soybean products for human consumption include; soy yogurt, soy milk and soy cheese [19, 24, 26]. Soy flour, weaning food formulations, soy based soups, confectioneries, beverages and fermented soy products as well as extruded products have also been documented [27]. Other known products are soy relishes, soy coffee, soy sausage and soy sprouts as well as tempeh, soy sauce, soy candies and soy meat [24, 28]. In Zambia Soybean is mainly being processed into cooking oil, animal feed and some products for human consumption such as soy flour, instant soy and soy pieces [8]. In spite of its food security potential, soybean cultivars generally contain the highest values of anti-nutritional factors compared to other legumes particularly Soybean agglutinin (SBA) which are lecithins with high affinity for galactose [29]. These can induce growth inhibition, cause pathological changes of intestinal tissue, and decrease in the immune system functioning [29]. Additional anti-nutrients include; trypsin inhibitors, phenols and phytic acid which together have strong mineral, protein and starch binding properties thereby decreasing the bioavailability of these nutrients [22, 30]. This calls for appropriate household processing techniques in order to prevent food- borne toxins and protect human health [29]. Some household techniques have been reported to reduce the anti-nutrients thus improving the rate and extent of starch and protein digestion in-vitro condition [31]. These include; soaking in 1% citric acid solution followed by cooking for 30 min, sprouting, boiling, pressure cooking and roasting [31]. At household level, it is recommended that several measures be applied at once as application of a single domestic processing method is insufficient for complete removal of anti-nutritional factors in soybean [30].

Processing and utilization of soybean for household consumption is influenced by a number of factors. Based on the Food Systems these are summarized as; socioeconomic, enabling environment and food environmental factors [5]. Socioeconomic factors include; awareness, production, market, farm diversification as well as membership to farmer's organization, age, and gender [5, 28, 32, 33]. Lack of knowledge on the health benefits of soybean and how to process it are the documented awareness factors [5, 28, 32, 33]. Farm size, poor soils, cost of improved soybean seeds, low fertilizer use and poor access to affordable credit services as well as poor farm diversification and depending on rain-fed agriculture are the production factors cited [9, 34–37]. Meanwhile, poor access to favourable soybean output and processing markets are the documented market related factors while occupation and gender are the additional socioeconomical factors [9, 32–39]. Enabling environmental factors include; yield of soybean, access to agricultural advisory services, household size as well as value chain governance and access to technology [5, 9, 24, 27, 28, 34, 36, 37]. Beany flavour with associated anti-nutritional factors as well as the hard-to-cook characteristics associated with soybean are some of the known food environmental factors [24, 32, 34–37, 40].

Soybean processing and utilization is still low in Eastern Province while high levels of malnutrition and food insecurity prevail despite being among the leading producers of the nutritious crop soybean [10, 41]. The province also receives support from international and local projects, such as the Green Innovation Centers (GIC), Food and Nutrition Security Enhanced

Resilience (FANSER) Projects under Deutsche Gesellschaft für Internationale Zusammenarbeit (GIZ), and, Catholic Relief Services (CRS). There is limited evidence on factors that influence household processing and utilization of soybean to inform formulation of strategies for processing and utilization of the legume in the Eastern Province of Zambia. Therefore, the main objective of this enquiry was to determine the best ways of improving household soybean processing and utilization in the Eastern Province of Zambia for possible adoption by the GIZ funded projects as well as other projects operating in similar environments. This included the determination of the extent of utilization, associated factors as well as construction of an enabling framework. To the best of our knowledge this is the first research in this regard in Zambia.

## Methodology

### Study setting

This study was conducted in the Eastern Province, one of Zambia's ten provinces. The province lies between Malawi on the east and Mozambique on the south [42]. Locally it shares borders with three other provinces of the country, namely, Lusaka, Central and Muchinga [42]. With the provincial capital being Chipata, eastern province has an area of 51,476 km2. A population of 1,592,661, accounting to 12.16% of the total Zambian population was recorded in 2010 with 1,030 for females for every 1,000 males [43]. Chewa was the largest community in the region and the most widely spoken language with 34.6 per cent people speaking it [43]. The study was conducted in the three districts. These were Petauke, Katete and Chipata where the predominant economic activity is farming and have a GIZ geographical foot print [43, 44]. Fig 1 shows the map of eastern province.

### Study design

In this study all data were collected at the same time and triangulated at results stage thereby generating rich information to inform programming. A concurrent triangulation study design was therefore appropriate [45]. Two other study designs were nested in this strategy. These were a cross sectional survey which was used to collect quantitative information as well as a barrier analysis which was used to generate in-depth data that provided additional insights.

### Cross sectional survey

A cross-sectional survey generated information on factors associated with Soybean processing and utilization using a structured questionnaire. A pilot study conducted in Petauke district to test the data collection instrument preceded the main study. The instrument was later improved on to ensure accuracy and reproducibility of the results. Up to 168 households participated in the pilot study. This was followed by questionnaire update in the kobo collect toolbox, which was redeployed in real time for accessibility and data collection immediately using smart phones. The questionnaire was finally administered completely on 1,237 households out of 1258 households planned for in the study giving a non-response rate of 1.67%. This rate is lower than that recorded (4%) in the one national household surveys [41].

**Sampling for the cross sectional survey.** In order to determine the sample size, the study adopted the Yamane (1967) technique, which states that the sample size n is defined as:

$$n = \frac{N}{1 + N(e)^2}$$

Where n is the sample size, N is the population size and e is the level of precision. At 95%

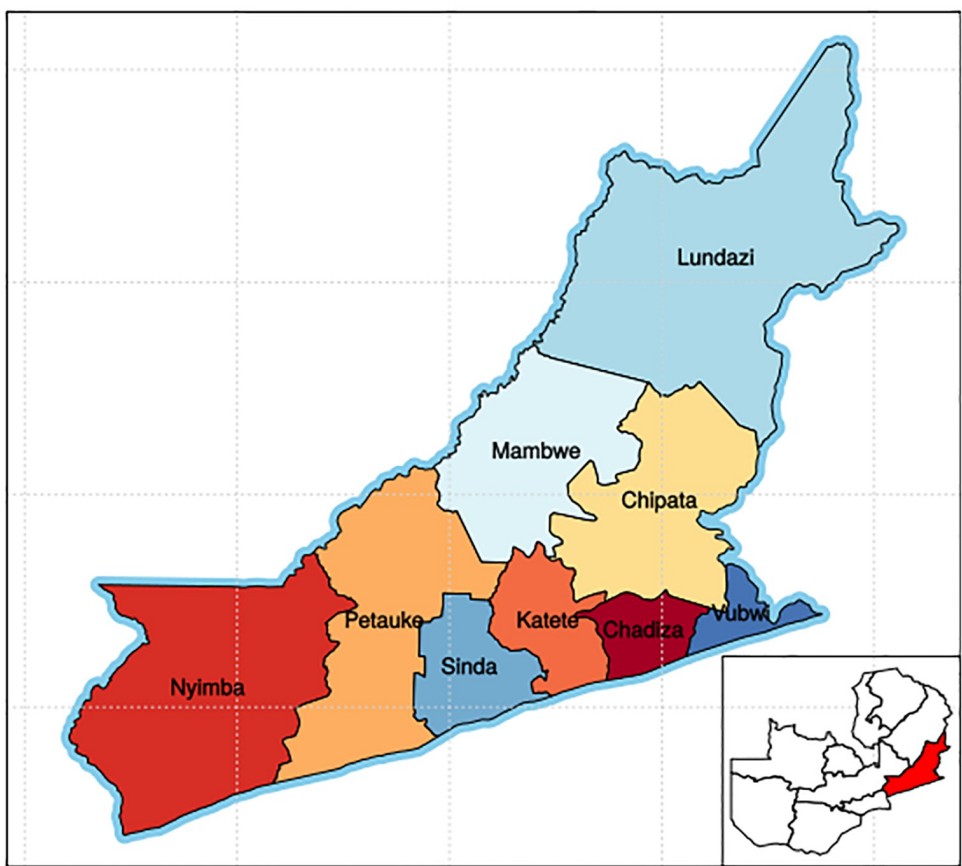

**Fig 1. Map of eastern province of Zambia.**

confidence level, e = 0.05 [46]. Yamane's technique was appropriate in this case as evidence on the rate of household whole soybean processing and utilization was limited. Yamane's technique therefore helped in calculating a representative sample size which produced results that reflect the true population picture and can be reproducible. The sample size was adjusted for non-response rate with the prediction of the rate adopted from the recent Zambia Demographic and Health Survey [41]. Each district had a specific sample size calculated based on their population size. The population figures for calculating sample sizes were obtained from the city population website [47].

**Selecting clusters for cross sectional survey.** In this study, five clusters from each district were included in the study (Fig 2). A cluster was defined as a ward, which is an official administrative unit under a district in Zambia. The number of wards selected were matching with five working days in each district. The wards were segmented using camps as boundaries manned by camp officers under the Ministry of Agriculture. This was in order to effectively collect the required data. The five clusters for each district were randomly assigned by probability proportional to size (PPS) using the ENA software [48]. The selection of the clusters was conducted at district level after excluding the wards that were based in town. One segment from each ward was selected according to PPS. The boundaries used for the segmentation were the agricultural camps manned by a camp officer in the Ministry of Agriculture. In each segment, PPS was also used to select villages. In cases where the targeted number of households was not reached, the adjacent village was included in the study. The PPS technique

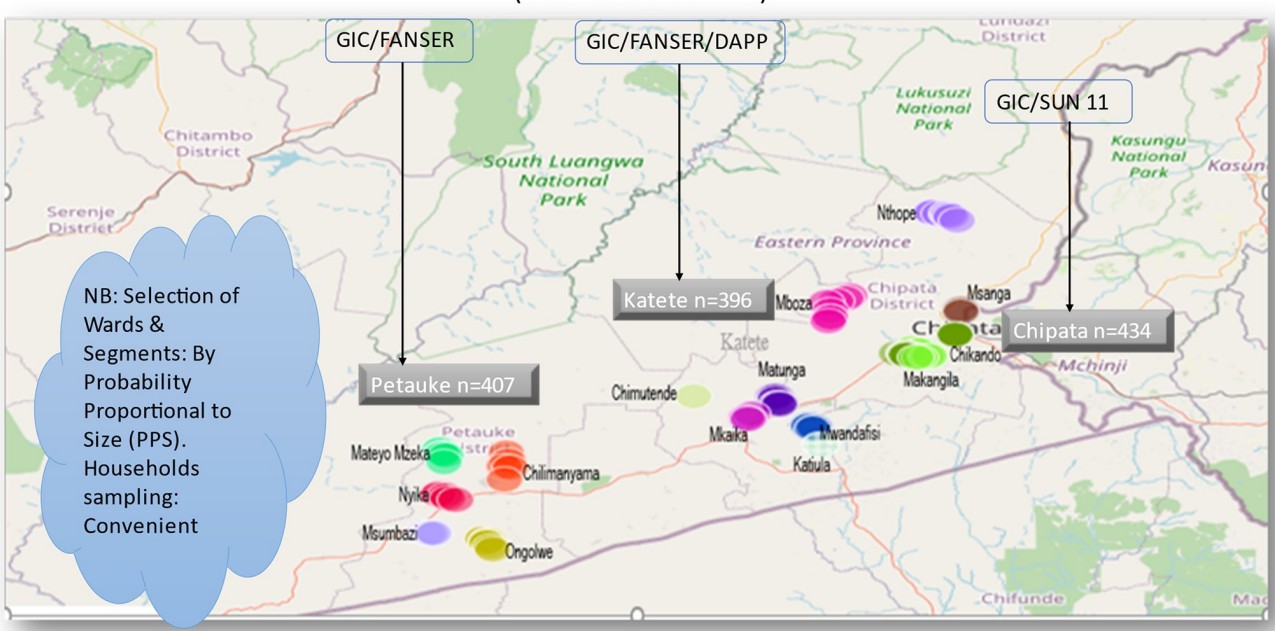

**Fig 2. Distribution of selected clusters.**

reduced the effect of random errors at sampling stage and helped get a representative sample that would yield a true district rate of household soybean utilization.

**Selecting households for cross sectional survey.** This study was conducted at the beginning of the planting season when most households went out every morning to prepare fields and gardens. Convenient sampling method was therefore conservative enough in this case. This meant that as soon as a household was available, they were recruited and interviewed. Some of the respondents were followed in their fields during demonstrations organized by agricultural extension officers. Informed consent was obtained from all before commencement of the interview. The participants were also informed that they were free to decide not to take part in the interview and that they were also free to with draw at any time. There were no risks for participants in this study. Participants were encouraged to participate as the findings were necessary to shape the best ways of processing and utilization of soybeans in households to improve on dietary diversity and ultimately nutrition status. These study specific ethical considerations are according to the Helsinki declaration of 1964 on studies involving human subjects [49].

**Training data collectors for cross sectional survey.** A one day training workshop for survey team members was conducted. The training covered general survey objectives, overview of survey design, household selection procedures, data collection and interview skills. In addition, the survey team members were closely monitored during data collection in the field. All this reduced on variations between data collectors thereby yielding precise and accurate results.

**Data analysis for cross sectional survey.** Quantitative data was downloaded from the kobo collect toolbox in the format suitable for excel sheet (XLS legacy). This was then cleaned and imported into the Stata software followed by coding and analysis. Frequencies were used to describe processing and utilization of whole and ready-made soybean products, household soybean production as well as some socioeconomic and sociodemographic characteristics of

participants. Mean (SD) was used to describe age. All variables were later fitted into the multiple logistic regression model to come up with the adjusted estimates in the most efficient model that rules out confounding factors. The variables both significant and those not significant at <0.05 were entered using weighted logistic regression. After controlling for all the other factors a number of them were found to be associated with soybean processing and utilization (p<0.05). All the factors associated with soybean processing and utilization were entered into the best fit (final) model in order to report Adjusted Odds Ratio with 95% Confidence Interval.

## Barrier analysis

Barrier Analysis was conducted by going through the procedure in Table 1. This is a Participatory Analysis Tool that identifies key enablers and barriers to the implementation of practices in resource-poor communities [50]. In order to generate information on the barriers or enablers for soya bean processing as well as utilization, Focus Group Discussions (FGDs) and In-depth Interviews (IDIs) functioned as data collection instruments with the help of semi structured question guides. FGDs were chosen in order to reveal collectively shaped social processes [51]. On the other hand In-depth Interviews were chosen to get a vital source of information being the perspectives of individuals who have personal experiences with Soybean processing and utilization [52, 53].

**Sampling for barrier analysis.** Community leaders were purposively sampled for Focus Group Discussions (FGDs). The number of FGDs was determined by theoretical saturation [54]. This means that participants for FGDs were recruited and Discussions were held until a point where there was no more additional information being generated [54]. In-depth interviews (IDIs) were equally conducted, with the key informants purposively sampled. These came from the Ministries of Health, Agriculture as well as from GIZ and CRS. Informed Consent was obtained from all participants before interviews. Each FGD was conducted by two moderators and digitally recorded. Permission to record the discussions digitally was sought from all participants. The discussions were moderated by one facilitator who also ensured that all the topics were covered in the interview guide. A note-taker assisted with recording both digitally and by writing which helped in determining emerging themes. Each FGD lasted for an average of an hour. FGD venues used were mainly meeting sites for farmers with Camp officers, which are open places away from houses. Two out of six FGDs were conducted in a room at a health facility as well as in a classroom at a primary school. All IDIs were conducted via telephone. This is because it was not possible to meet participants physically due to distances as well as busy work schedules.

**Table 1. Conducting barrier analysis.**

| Stages of analysis | Methods and tools used |
|---|---|
| Identification of community leaders | In collaboration with Agricultural camp officers, Senior Headmen and Lead Farmers in the Wards, the headmen were identified |
| Identification of potential barriers | Extensive literature review, brainstorming with the community leaders during FGDs and meeting with key informants during IDIs |
| Analysis of barriers | Detailed analysis using based on the Food Systems Conceptual Framework |
| Screening and Validation of important barriers and measures | Validation through presentation |
| Submission of Draft and Final Report | Submission |

Adopted from: Kittle, 2017

**Data appraisal for barrier analysis.** Audio recorded data was transcribed verbatim and merged with the notes that were taken. This was followed by reading transcripts repeatedly in order to gain a deeper insight of the data [55]. Coding was the next stage. A code is a word, sentence or phrase that represents aspects, captures the essence or features of a data [56]. Codes were then matched with the food systems thematic areas. These are socioeconomic factors, factors in the enabling and food environment [57]. With the original meaning of what was communicated by the informants maintained coding was carried out using NVIVO QSR10 (QSRInt, Melbourne Australia).

## Ethical considerations

This research was approved by the Levy Mwanawasa Medical University Research Ethics Committee (LMMU-REC 00010/20) as well as the National Health Research Authority (NHRA). Permission was obtained from Petauke, Katete, Chipata, Chipangali as well as Kasenengwa district agricultural coordinators' offices. Permission to collect data was also obtained from the senior headmen in each ward. Since this was a low risk research informed consent was verbally obtained from respondents. Respondents were selected and interviewed in their homestead away from other family members in order to ensure privacy. The respondents were free to withdraw from the study at any time. Data collected was de-identified.

## Results and discussion

### Characteristics of participants

**Sociodemographic characteristics of cross sectional survey respondents.** Up to 1,237 out of 1258 households participated in the study giving a non-response rate of 1.67%. The mean age of respondents was 40.33(SD = 13.63) years. Male headed households were 780/1237 (63%) being more than female headed households which were 457/1237(37%). National values for male headed households are at 74.2% slightly more than what was reported in this study. This however shows that male headed households are more in Zambia compared to female headed households (CSO, 2018). Eastern Province is also known to have more females than males. This is confirmed by the sex ratio of 2010 census which was 1,030 for every 1,000 males (CSO, 2010). Despite all this up to 733/1,237(59.26%) females more than males [504/1,237 (40.74%)] participated in the Cross Sectional Survey. This could mean that females were found in the households during the survey while the males were mostly absent. It is not clear from this study where the men were at the time of the survey. On the other hand, men 475/1,180 (40.25%) reported less involvement in general farming activities than 705/1,180 (59.75%) females. Meanwhile, males up to 23/46 (50%) and females up to 23/46 (50%) were equally involved in off farm business activities. The majority [669/1,237(54.08%)] of the participants were Chewa speaking. These were found in substantial numbers in all the three districts and comprised almost the entire population [387/396(97.73%)] in Katete District. This is in agreement with the 2010 census whereby Chewa was found to be the largest community in Eastern Province with 39.7 per cent of the total population and Chewa was the most widely spoken language with 34.6 per cent speaking it (CSO, 2010). Table 2 shows the details.

**Socioeconomic status of cross sectional survey respondents.** Primary education [431/ 1237(34.84%)] was the highest education level reported by most respondents. Petauke [173/ 1237(42.51%)] had the highest proportion with Katete [88/1237(22.22%)] trailing behind. Katete also had a number of respondents with some primary, as well as, no education at all [126/396(31.82%)] and [110/396(27.78%)] respectively. This agrees with the Second Report of the Committee on Education, Science and Technology for the Fourth Session of the Tenth National Assembly appointed of 24 September 2009, which showed that among the districts

**Table 2. Sociodemographic characteristics.**

| Variable | Category | Total (N = 1237) | Petauke (n = 407) | Katete (n = 396) | Chipata (n = 434) |
|---|---|---|---|---|---|
| Household head | | | | | |
| Gender n (%) | Males | 780(63.06) | 230(56.51) | 306(77.27) | 244(56.22) |
| | Females | 457(36.94) | 177(43.49) | 90(22.73) | 190(43.78) |
| Respondent | | | | | |
| Age Mean (SD) | | 40.33(13.63) | 40.65(11.42) | 40.09(14.17) | 40.25(14.99) |
| Gender n (%) | Males | 504(40.74) | 197(48.40) | 149(37.63) | 158(36.41) |
| | Females | 733(59.26) | 210(51.60) | 247(62.37) | 276(63.59) |
| Ethnic affiliation n (%) | Chewa | 669(54.08) | 161(39.56) | 387(97.73) | 121(27.88) |
| | Ngoni | 295(23.85) | 2(0.49) | 5(1.26) | 288(66.36) |
| | Nsenga | 244(19.73) | 237(58.23) | 2(0.51) | 5(1.15) |
| | Tumbuka | 12(0.97) | 2(0.49) | 1(0.25) | 9(2.07) |
| | Other | 17(1.37) | 7(1.72) | 0(0.00) | 11(2.53) |

with unacceptable levels of adult illiteracy, Katete was the highest with 62.9%. Farming [1,180/ 1237(95.39%)] was the predominant occupation reported in the three districts. Some off farm business activities were documented in Petauke [34/407(8.35%)] and Chipata [10/434 (2.30%)]. Males [23/46 (50%)] and females [23/46 (50%)] controlled these activities equally. Mean household size was 6(SD = 2) slightly higher than the national value of 5.2 reported for rural settings in the 2018 Zambia Demographic and Health Survey (CSO, 2018). Four major consumer goods owned in the three districts include a working mobile telephone, a working radio, owning a bed and having electricity (MTN set or solar panel) with 868/1237(70.17%), 431/1237(34.84%), 424/1237(34.28%), 391/1237(31.61) respectively (Table 3).

**Socio-demographic characteristics of barrier analysis participants.** There was a total of six focus group discussions (FGDs) conducted in this study with two FGDs carried out in each one of the selected wards in the three study districts. Up to 36 participants with median age of 50(Range = 28–75) years participated. The youngest participant as well as the oldest participant aged 28 and 75 years respectively were from Chipata district. In-depth Interview (IDI) participants were six. Their median age was 41(Range = 36–57) years. Table 4 highlights the details.

## Soybean production, processing and utilization

**Production.** The overall proportion of respondents who reported growing Soybean was 668/1237(54%). Soybean growing was reported more in Chipata [332/434(79.49%)] followed by Katete [280/396(70.70%)]. Reports of soybean growing in Petauke were only 56/407 (13.76%). Perceptions of headmen showed that soybean growing varied by district. Nthope ward in Chipangali, a new district, which fell under Chipata in this study was ranked first. While, Chimutende ward in Katete was ranked second. Meanwhile, Makangila ward in Chipata as well as Katiula ward in Katete were ranked third. On the other hand, Chilimanyama ward in Petauke district was ranked fourth. There were limited reports of growing soybean in Msumbazi ward in Petauke district. These findings are consistent with a report by Lubungu and colleagues which show that soybean production in eastern province is concentrated in Lundazi, Chipata, and Chadiza with erratic participation reported in Katete and Nyimba (Lubungu et al., 2013). This study has also shown that soybeans growing reports were massive among the male headed households [466/668 (70%)] than among the female headed households [202/668 (30%)]. The incidence of adoption among the female-led households is low possibly because they are constrained by lack of access to input, credit, and extension services (Hossain, 2019). Details are shown in Table 5.

**Table 3. Socioeconomic status.**

| Variable | Category | Total (N = 1237) | Petauke (n = 407) | Katete (n = 396) | Chipata (n = 434) |
|---|---|---|---|---|---|
| Respondent | | | | | |
| Highest education level n (%) | Secondary | 42(3.40) | 5(1.23) | 17(4.29) | 20(4.61) |
| | Some secondary | 210(16.98) | 71(17.45) | 55(13.89) | 84(19.36) |
| | Primary | 431(34.84) | 173(42.51) | 88(22.22) | 170(39.17) |
| | Some Primary | 337(27.24) | 110(27.03) | 126(31.82) | 101(20.74) |
| | None | 217(17.54) | 48(11.79) | 110(27.78) | 59(13.59) |
| Occupation n (%) | Farmer | 1,180(96.39) | 370(90.91) | 391(98.74) | 419(96.54) |
| | Farmer/business | 46(3.72) | 34(8.35) | 2(0.51) | 10(2.30) |
| | Other* | 11(0.89) | 3(0.74) | 3(0.76) | 5(1.15) |
| Household size mean(SD) | | 6(2.00) | 6(2.00) | 6(3.00) | 6(3.00) |
| Own a bed | Yes | 424(34.28) | 138(33.91) | 150(37.88) | 136(31.34) |
| | No | 813(65.72) | 269(66.09) | 246(62.12) | 298(68.66) |
| Own electricity | Yes | 391(31.61) | 66(16.22) | 146(36.87) | 179(41.24) |
| | No | 846(68.39) | 341(83.78) | 250(63.13) | 255(58.76) |
| Own working refrigerator | Yes | 29(2.34) | 5(1.23) | 18(4.56) | 6(1.38) |
| | No | 1,208(97.66) | 402(98.77) | 378(95.45) | 428(98.62) |
| Own a working television | Yes | 121(9.78) | 25(6.14) | 43(10.86) | 53(12.21) |
| | No | 1,116(90.22) | 382(93.86) | 353(89.14) | 381(87.79) |
| Own a working radio | Yes | 431(34.84) | 138(33.91) | 112(28.28) | 181(41.71) |
| | No | 806(66.16) | 269(66.09) | 284(71.72) | 253(58.29) |
| Own a working telephone | Yes | 868(70.17) | 310(76.17) | 238(60.10) | 320(73.73) |
| | No | 369(29.83) | 97(23.83) | 158(39.90) | 114(26.27) |
| Grow soybean | Yes | 668(54.00) | 56(13.76) | 280(70.70) | 332(79.49) |
| | No | 569(46.00) | 351(86.24) | 116(29.29) | 102(23.50) |
| Grow soybean & Owned a bed | Yes | 274(41.02) | 37(66.07) | 127(45.36) | 110(33.13) |
| | No | 394(58.98) | 19(33.93) | 153(54.64) | 222(66.87) |

*Other occupation such as business, combines farming with working, doing nothing

## Processing and utilization

**Utilization of ready-made soybean products.** A Textured soya protein (TSP) locally known as soya pieces or nyama soya was the only ready-made product reported to be regularly consumed in the rural areas in the three districts. It was universally consumed [1,030/1237 (83%)] as an easily accessible and relatively affordable relish to accompany a staple cereal thick porridge (nshima). Varying consumption levels were reported being more in Petauke [373/ 407(92%)] followed by Katete and Chipata with 321/396 (81%) and 334/434(77%) respectively (Table 6). TSP was congruent with the dietary pattern in the study districts. One participant stated;

"On a daily basis we eat nshima or samp, whole soybean is known for money we just leave seed, we only consume it in the form of soya pieces"

(A Headman in Katete district, FGD002)

In this study the least producer of soybeans, Petauke district emerged the highest consumer of the ready-made soybean product the TSP, with households spending up to USD15 per

**Table 4. Barrier analysis participants.**

| District | Planned number Participants | Actual Number of Participants | Gender | | Ward/Organization | Data Collection Method | Age Range |
|---|---|---|---|---|---|---|---|
| | | | Male | Female | | | |
| Petauke | 20 | 16 | 14 | 2 | Chilimanyama | FGD* | 32–72 |
| | | | | | Msumbazi | | |
| | 2 | 2 | 2 | 0 | MoA[a] | IDI** | 36–57 |
| | | | | | COMACO[b] | | |
| Katete | 20 | 8 | 7 | 1 | Katiula | FGD* | 31–67 |
| | | | | | Chimtende | | |
| Chipata | 20 | 12 | 11 | 1 | Nthope | FGD* | 28–75 |
| | | | | | Makangila | | |
| | 4 | 4 | 2 | 2 | MoA, FANSER[c], GIC[d] & GIZ[e] | IDI** | 38–47 |

[a]Ministry of Agriculture;

[b]Community Markets for Conservation Ltd;

[c]Food and Nutrition Security Enhanced Resilience;

[d]Green Innovation Centers;

[e]Deutsche Gesellschaft für Internationale Zusammenarbeit;

*Focus Group Discussion;

**In-depth Interview

**Table 5. Soybean production.**

| | Total N = 1,237 | Petauke n = 407 | Katete n = 396 | Chipata n = 434 |
|---|---|---|---|---|
| [a]HH Soybean growing n (%) | 668/1,237(54.00) | 56/407(13.76) | 280/396(70.70) | 332/434(79.49) |
| Male headed | 466/668(70.00) | 38/56(67.86) | 227/280(81.07) | 201/332(60.54) |
| Female headed | 202/668(30.00) | 18/56(32.14) | 58/280(20.71) | 131/332(39.46) |
| In HH that own a bed | 274/668(41.02) | 37/56(66.07) | 127/280(45.36) | 110/332(33.13) |

[a]Households

**Table 6. Utilization of textured soya protein.**

| | Total N = 1,237 | Petauke n = 407 | Katete n = 396 | Chipata n = 434 |
|---|---|---|---|---|
| [a]HH Soybean utilization n (%) | 1,030/1,237(83) | 373/407(92) | 321/396(81.00) | 334/434(77.00) |
| [b]TSP | 1,030/1,237(83) | 373/407(92) | 321/396(81.00) | 334/434(77.00) |
| [b]TSP Budget <USD6/month | 871/1,030(81.02) | 316/375(82.72) | 257/321(77.88) | 298/334(82.09) |

[a]Households,

[b]Textured Soya Protein such as soy chunks, soy pieces, USD: United States Dollar

month. Households buy the product from the local retail outlets. This demonstrates the willingness to buy one of the ready-made commercial soybean products. In addition it also shows that communities with disposable income such as those from Petauke with some off farm activities as reported in this study were able to buy ready-made soybean products.

It is also worth noting that most of the TSP found during the study period were from Malawi (Fig 3). The local shop owners however confirmed that they occasionally stocked one local brand. This could mean an unmet market gap by the local soybean value addition

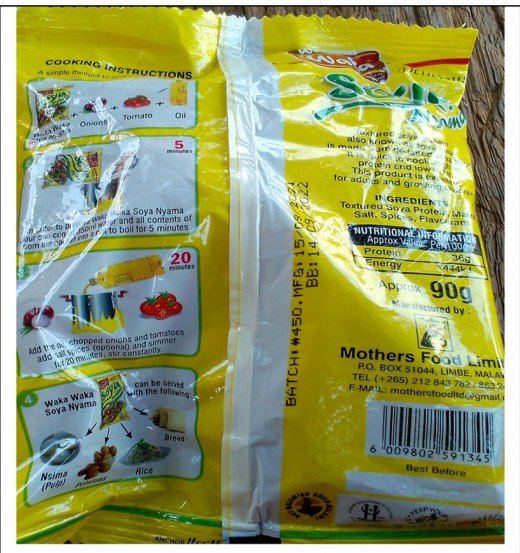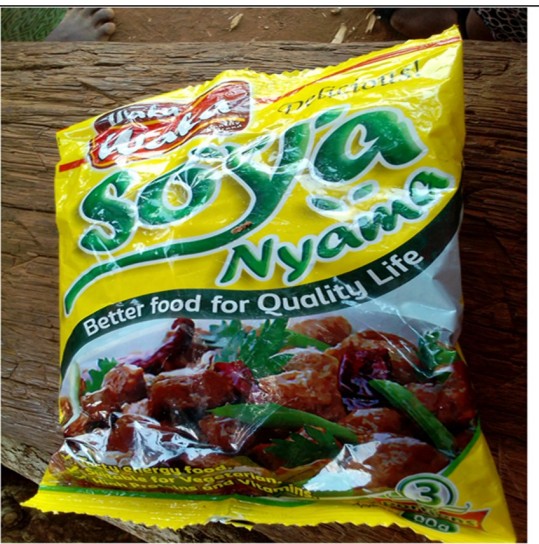

**Fig 3. Textured soya protein.**

companies. Raw soybean being exported to Malawi during the significant amount of trade in soybeans taking place across the Zambia-Malawi border [58] is processed there coming back as TSP. One of the companies that locally make readymade soybean products is the Community Markets for Conservation (COMACO) a non-profit private company. The company is located in Chipata the provincial headquarters of Eastern Province with a branch in Petauke district. Soybean products manufactured by COMACO include yummy soy, crude oil and TSP. These are distributed in supermarkets and other retail outlets as well as to NGOs involved in school feeding programmes [10].

**Processing and utilization of whole soybean.** In Table 7 almost half [589/1237(48%)] of the respondents reported processing and utilizing whole soybean in family meals. Chipata ranked the highest [260/434(60%)], followed by Katete [179/396(45%)] and Petauke [150/407

**Table 7. Utilization of whole soybean.**

| | Total N = 1,237 | Petauke n = 407 | Katete n = 396 | Chipata n = 434 |
|---|---|---|---|---|
| [a]HH Soybean utilization n (%) | 1,030/1,237(83) | 373/407(92) | 321/396(81.00) | 334/434(77.00) |
| Whole Soybean | 589/1,237(48.00) | 150/407(37.00) | 179/396(45.00) | 260/434(60.00) |
| Male headed | 381/589(64.69) | 81/150(54.00) | 146/179(81.56) | 154/260(59.23) |
| Female headed | 208/589(35.31) | 69/150(46.00) | 33/179(18.44) | 106/260(40.77) |
| Male respondent | 249/589(42.28) | 73/150(48.67) | 68/179(37.99) | 108/260(41.54) |
| Female respondent | 340/589(57.73) | 77/150(51.33) | 111/179(62.01) | 152/260(58.46) |
| Grow Soybean | 426/668(63.77) | 44/56(78.57) | 155/280(55.36) | 227/332(68.37) |
| Owned a bed | 243/589(41.26) | 52/150(34.67) | 93/179(51.96) | 98/260(37.69) |
| Correct processing | 479/589(81.32) | 127/150(84.67) | 156/179(87.15) | 196/260(75.38) |
| Products | | | | |
| Porridge | 279/589(47.37) | 35/150(23.33) | 102/179(56.98) | 141/260(54.23) |
| Soy Milk | 68/589(11.54) | 9/150(6.00) | 39/179(21.79) | 20/260(7.69) |
| Confectionaries | 35/589(5.94) | 2/150(1.33) | 2/179(1.12) | 31/260(11.92) |

[a]Households

(37%)]. The differences in soybean processing and utilization reports among the three districts were significant (p<0.0001) with Petauke explaining much of the difference. Notably, soybean utilization is high where soybean is produced more. However, accessibility of whole soybeans for household consumption throughout the year was negligible [3/1030(0.29%)] even among households that grew it. This means that the households were only able to make various products during the harvest season. Utilization of whole soybeans in family meals could bring about improvement food security and nutritional status in households [15–17].

*Whole soybean products at household level.* Soybean porridge was the most frequently reported whole soy dish [279/1237 (22.55%)] with higher proportions in Chipata [141/434 (32.49%)] followed by Katete [103/396(26.01%)] and Petauke [35/407(8.60%)]. The porridge was prepared out of the maize-soybean blend milled together. Soybean used was either heat treated or raw. The milled product was sometimes used to make nshima and confectionary products to a limited extent. Cakes made from the blend were also known as 'Vigumu' in local language. Fritters were equally made from the soy flour mixed with wheat flour. These were reported to be sometimes sold within the neighbourhood. Other products reported to a limited extent include boiled whole soybeans, soy sausage and soy coffee as well as soy milk (Table 7). The soybean products could also be fortified with animal source foods such as fish powders to make them nutrient dense [59, 60].

*Correct whole soybean processing method reports.* Table 7 also shows households that reported correct whole soybean processing methods. Up to 479/724 (66%) reported having correctly processed whole soybeans by preheating it with dry heat before mixing it with maize in preparation for milling using a harmer-mill. There were more reports of correct whole soybean processing from Petauke [127/134(95%)] than from Chipata [196/318 (62%)] and Katete [196/318 (62%)]. Consumption of raw whole soybean is associated with adverse nutritional effects due to the presence of endogenous inhibitors of digestive enzymes and lectins causing poor digestibility [61]. Nutritional quality of soy foods can be improved by inactivating inhibitors and lectins using heat treatment [61]. Dry heat treatment can be applied at high temperatures above 120°C [62]. In a rural setting this could be achieved by roasting the dry cleaned soybean to slightly brown colour.

**Factors associated with soybeans processing and utilization at household level.** Table 8 shows a multivariate analysis. All the variables were first fitted in the multiple logistic regression model to come up with the adjusted estimates in the most efficient model that rules out confounding factors. The variables both significant and those not significant at <0.05 were entered using weighted logistic regression. After controlling for all the other factors a number

Table 8. Factors associated with whole soybean processing and utilization.

| Variable | Multivariate Analysis | |
|---|---|---|
| | *AOR (95%CI) | P-Value |
| District | 0.76(0.58 0.98) | 0.038 |
| Gender of Household head | 1.94(1.21 3.13) | 0.006 |
| Gender of respondent | 0.47(0.30 0.73) | 0.001 |
| Ethnic affiliation | 1.16(1.08 1.25) | 0.000 |
| Owning a bed | 1.75(1.22 2.49) | 0.002 |
| Growing Soybean | 4.47(2.82 7.08) | 0.000 |
| Preparing Porridge/Nshima | 816.37(110.83 6013.31) | 0.000 |
| Processing barrier | 0.42 (0.28 0.62) | 0.000 |

*Adjusted Odds Ratio

of them were found to be associated with soybean processing and utilization (p<0.05). These are discussed in the sections that follow.

**Factors that increase chances of soybean processing and utilization at household level.** Factors that positively influenced soybean processing and utilization include; belonging to a male headed household AOR 2.15; 95% CI 1.21 to 3.79, the Chewa community AOR 1.26; 95%CI 1.15 to 1.38 and higher social hierarchy indicated by owning a bed AOR 1.78; 95%CI 1.16 to 2.74. In addition reporting preparing porridge or Nshima AOR 1905.14; 95%CI 241.50 to 15029.39, as well as growing soybeans AOR 11.92; 95%CI 5.34 to 26.60 were equally associated with increased chances of soybean processing and utilization ([Table 8]). Male headed households were reported to grow more soybean (70%) than those headed by females (30%). Chewa speaking people had increased chances because they are the largest community in Eastern Province with 39.7 per cent of the total population as at 2010 population census. They are almost the only ethnic group in Katete district (97.73%) which emerged second in soybean processing and utilization in this study. Households who owned a bed had increased chances of processing and utilizing soybean as this was a sign of higher social hierarchy in a rural setting [63]. This could have been brought about by agricultural activities particularly growing soybeans as a cash crop. In this study, households that owned a bed as well as grew soybeans were not significantly different from those that owned a bed and utilized soybean p = 0.213. This means that the proportion of households that reported owning beds was similar to the proportion that reported growing and utilised whole soybean. Among the Soybean products, preparing porridge was one product that increased the chances of soybean processing and utilization AOR 866.51; 95%CI 116.32 to 6454.96. This entails that porridge or nshima are the whole soybean products that most households (23%) attempted to make ([Fig 4]). In this case, whole soybeans was milled and mixed with maize and prepared as porridge or nshima. This could indicate that these products were harmonious with meal patterns. FGDs with headmen confirmed this by showing that porridge and nshima were common staple meals thereby making it easy to adopt utilization of soybeans through these products. Once available, the composite flour (mixture of maize and soybean) was at times used to make soybean cake (Vigumu in local language). Soybean utilization improves on dietary diversity in households particularly for mothers and children during the 1000 most critical days in life when optimal linear growth occurs in children [64]. Utilization of protein from a legume such as soybean in the human body require that all essential amino acids are present in amounts that can promote health [65]. Soybean however is limiting in sulphur containing amino acids, methionine and cysteine [65]. Methionine is the initiating amino acid in the synthesis of virtually all eukaryotic proteins [65]. Cysteine, by virtue of its ability to form disulphide bonds, plays a crucial role in protein structure and in protein-folding pathways [65]. Soybean should therefore be combined with whole cereals to ensure efficient protein utilization. Cereals for combining with soybean

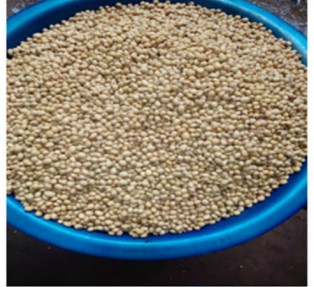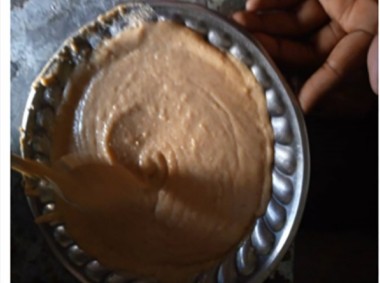

**Fig 4. Whole soybean & soy porridge.**

should be milled whole in order to maintain amino acid content and avoid loss during bran and germ removal [66].

**Factors that reduce chances of soya bean processing and utilization at household level.** Study district AOR 0.33; 95%CI 0.22 to 0.50, gender particularly being a male respondent AOR 0.41; 95%CI 0.24 to 0.72, as well as experiencing soybean processing barrier, AOR 0.42; 95%CI 0.26 to 0.69 reduced the chances of soybean processing and utilization (Table 8). Processing and utilization of whole soybeans was generally low in all the three districts. Petauke had the lowest prevalence being 150/589 (25%), followed by Katete with 179/589 (30%) and Chipata with 260/589 (44%). Meanwhile, only up to 249/1237 (20%) of the male respondents reported utilizing whole soybeans. It should also be noted that lack of knowledge in soybean use was reported more among the male respondents as compared to females during the FGDs with the headmen. Men are not conversant as they are not primarily involved in cooking or preparing food especially if there is no money involved [34]. Men know soybean as a cash crop. This is despite the deliberate strategies in place by different organizations to include men during implementation at household level. There is need to re-examine the male involvement approach. A headman in Katete district during FGD004 said ". . .men are not involved in soybean processing and utilization. If a man is not involved he will just want to sell". On the other hand up to 670/1237 (54%) households experienced whole soybean utilization barriers. Respondents who did not utilize soybean for household consumption mainly cited lack of knowledge (55.655%) followed by lack of soybean (33%). This makes the two reasons potential activities for targeting to improve soybean utilization (Table 8).

## Barrier analysis findings

**Emerging themes based on the food systems.** Based on the food systems, a number of factors influencing soybean processing and utilization emerged. Socioeconomic factors that were reported include; awareness, gender issues, age, market, land, soil and policy. While enabling environmental factors itemised include; availability of soybean, availability of ingredients and accessibility of soybean, extension services, technology as well as value chain governance. On the other hand food environmental factors recounted include; availability of a ready-made TSP) soybean product as well as whole soybean products aroma, convenience and texture (Table 9).

## Enabling framework

**Food environment.** In the food environment, soybean distribution stage particularly; storage and processing as well as consumption stage should be targeted through various activities at household level. Storage of soybean for household use should be encouraged. Households that cannot grow it should be encouraged to buy and store it for household use. They should also be encouraged to buy processed soybean products. In addition, off farm business activities should be encouraged in households in order to help them generate disposable income which they can use to finance soybean production or buy nutritious soybean products. Meanwhile, consumption of TSP as a plant based low cost meat alternative and preparation of porridge or nshima from milled soybean-maize blend should be escalated. Inadequate processing methods reported with possible presence of anti-nutrients, reports of coarse milled soybean as well as hard to cook and labour intensity characteristics of soybean should be targeted for improvement (Table 10).

**Enabling environment.** In the enabling environment soybean production, distribution, acquisition and consumption stages are the possible boulevards for enhancing soybean processing and utilization at household level. Specific undertakings should include; improving

**Table 9. Processing and utilization of whole soybean-emerging themes.**

| Food Systems Category/Major barrier | Sub-barrier | | |
|---|---|---|---|
| **Socioeconomic** | | | |
| Awareness | Lack of knowledge was cited in about 40% of the households as contributing factor for Soybean processing and utilization. | | |
| | There was also lack of awareness about health and nutritional benefits | | |
| | Soybean processing at household level was unstandardized | | |
| Gender | Low male involvement in processing and utilization of soybean Low participation of female headed households in soybean production and therefore processing and utilization | | |
| Age | Low participation of households headed by the old aged is soybean production and processing | | |
| Market | Lack or unreliable soybean outputs and soybean processing markets in some wards to enhance market led growth and utilization | | |
| Land and soil | Lack of land and poor soils negatively affect soybean production and processing | | |
| Policy | Lack of government control on soybean marketing | Policy | Lack of government control on soybean marketing |
| **Enabling environment** | | | |
| Availability of Soybean | Lack of soybean as a results of low yield due to poor access to inputs, cost of improved seed, inoculum and fertilizer | | |
| | Lack of soybean due to poor access to affordable credit services | | |
| | Lack of soybean due to lack of postharvest storage. | | |
| Availability of ingredients | Lack of ingredients particularly for making confectionary products and soy sausage. | | |
| Accessibility of Soybean | Accessibility of whole soybeans for consumption throughout the year even among those that produced a lot of soybean particularly male headed household was not guaranteed. This may be due to lack of information on the benefits and how to process soybean. Utilisation was reported to last only during the harvesting season | | |
| Extension services | Inadequate coverage of existing projects & extension services on soybean processing and utilization. | **Extension services** | Inadequate coverage of existing projects & extension services on soybean processing and utilization. |
| | There were reports of some households not belonging to any organization that promoted production, processing and utilization of soybean | | |
| Technology | Poor access to technologies to mechanize production, harvesting and processing. | **Technology** | Poor access to technologies to mechanize production, harvesting and processing. |
| Value chain governance | Poor relationships among buyers, sellers, service providers and regulatory institutions that influence a range of activities required bring soybean from production to the end use (consumption). | **Value chain governance** | Poor soybean value chain governance |
| **Food environment** | | | |
| **Ready-made soybean products** | | | |
| TSP locally known as soya pieces | TSP was reported to be consumed by most households. They said; "This is our meat". All the TSP found in local shops was imported from the neighbouring country Malawi indicating a shortfall in local processing and supply. | | |
| **Whole soybean processing and utilization** | | | |
| Aroma | Inadequate processing methods reported in some households with characteristic beany flavour, possible presence of anti-nutrients and indigestion. | | |
| Convenience | Hard to cook characteristics especially when cooked whole. Labour intensity and time consuming with associated preparation neglect especially among smaller sized households when preparing some soybean products. | | |
| Texture | Coarse milled soybean | | |

**Table 10. Enabling framework.**

| Food Systems Category | Status | Strategy | Stage |
|---|---|---|---|
| **Food environment** | | | |
| **Ready-made soybean products** | | | |
| TSP locally known as soya pieces | Universally consumed mainly imported from Malawi | Encourage more local private sector players to venture into manufacturing of TSP to fill in the supply gap | Soybean distribution |
| **Whole soybean** | | | |
| Milled soybean-maize blend | Increased the chances of soybean processing and utilization | Milling of soybean and storage of soy flour in airtight containers for various uses especially fortification of cereal-based products should be enhanced. | Soybean distribution & consumption |
| *Beany flavour, possible presence of anti-nutrients* | Inadequate processing methods | Intensify cooking demonstrations applying more than one treatment method. | Soybean distribution & consumption |
| *Texture of the soybean maize meal blends* | Reports of coarse milled soybean | Encourage fine milling of soy flour using a 0.425mm mesh size or by repeat milling the soy flour using the hammer-mill mesh size available. | Soybean distribution & consumption |
| Convenience of the whole soybean | Hard to cook characteristics of soybean, labour intensity and time consuming | | |

access to inputs, affordable credit and soybean for consumption throughout the year as well as promotion of fortification of locally available foods with soybean products. Meanwhile, coverage of existing projects and extension services should be augmented. Furthermore, access to technologies, relationships among the buyers, sellers, service providers and regulatory institutions in the soybean value chain governance should be improved (Table 11).

**Table 11. Enabling framework.**

| Food Systems Category | Status | Strategy | Stage |
|---|---|---|---|
| **Enabling environment** | | | |
| Availability of Soybean | Lack of soybean. Poor access to inputs, cost of improved seed, inoculum and fertilizer | Enhance access to inputs. Farmers themselves should be involved in seed multiplication. | Soybean production |
| | Poor access to affordable credit services | Facilitate access to affordable credit facilities. | Soybean production |
| | Lack of postharvest storage. | Encourage post-harvest storage of some soybean for future use. Involve males in post-harvest storage | Soybean distribution |
| Availability of ingredients | Lack of ingredients for making some dishes like soy sausage | Encourage utilization of whole soybean in household recipes using locally available foods | Soybean consumption |
| Accessibility of Soybean | Accessibility of whole soybeans for consumption throughout the year was not guaranteed. | Encourage postharvest storage for future household consumption while involving males. Intensify cooking demonstrations fortifying local recipes while testing acceptability to bring about behavioural change in soybean utilization | Soybean distribution & consumption |
| Extension services | Inadequate coverage of existing projects & extension services on soybean processing and utilization. In some wards the program was new. | Promote collective action and full scale community involvement and ownership, capacity building and training, market development, creation of new products, development of cottage industries, incorporation of soybean into local dishes and diets, information exchange, trade and credit facilities, subsidized mineral fertilizers, import substitution agreement | Soybean production, distribution & consumption |
| Production & preparation neglect | Production and preparation neglect among small households | Promote convenient processing methods such as milling and storing for future use. Promote incorporation of soybean into local dishes and diets | Production, distribution & consumption |
| Technology | Poor access to production, harvesting and processing technologies. | Encourage privately owned equipment processing soybean in the districts. | Soybean distribution |
| Soybean value chain governance | Poor relationships among the buyers, sellers, service providers and regulatory institutions in the soybean value chain | Promote private initiatives Promote groups and associations, improve knowledge and information, Promote public-private partnerships | Production, distribution, acquisition & consumption |

**Socioeconomic factors.** Production, distribution, and acquisition as well as consumption stages should equally be targeted here. Soybean production should be hastened to improve utilization in all the districts among all ethnic groups. This should go hand in hand with market development, increasing processing and utilization awareness levels, involvement of male folks as well as female headed households. Wards with lack of land and poor soils should be target markets for soybean products. Time saving and low labour intensity correct processing methods such as heat treating, milling and correct storage of soybean should be promoted. Social protection should be escalated and made nutrition sensitive in order for the vulnerable groups to be able to access and utilize soybean. Government should be lobbied to recognize soybean in the local food system and enhance its inclusion in the Farmer Input Support Programme (FISP) Pack or any other future support programme to small holder farmers. Some strategies included in this study have also been reported elsewhere [30, 38, 67–70]. Table 12 summarises the findings.

## Conclusions

The purpose of this enquiry was to determine the best ways of improving soybean processing and utilization among households in the Eastern province of Zambia for adoption by the GIZ funded projects as well as other projects operating in similar environments. The results revealed that soybean processing and utilization was low in all the study districts. This was influenced by an interplay of factors. These were summarized as food environment, enabling environment as well as socioeconomic factors strewn through soybean production to consumption. In the food environment; consumption of TSP and preparation of porridge or nshima from milled soybean-maize blend positively influenced soybean utilization. Whereas beany flavour with associated ant nutrients, processing labour intensity, low accessibility of soybean for household consumption as well as lack of ingredients negatively influenced soybean utilization. In the enabling environment; lack of inputs, poor access to credit and inadequate coverage of existing projects as well as poor access to technologies were reported as impeding factors. Among the Socioeconomic factors, a higher social hierarchy, belonging to the Chewa community as well as off farm income and livestock ownership boosted soybean utilization. Meanwhile, lack of knowledge, low involvement of the male folks and female headed households were deleterious factors. Furthermore, low participation of the old aged, time and household size constraints were additional adverse socioeconomic factors. Likewise, unreliable soybean output markets, land and poor soils in some wards as well poor soybean value chain governance were auxiliary negative factors.

## Recommendations and priorities

Immediately in the food environment, there is need to enhance milling of soybean and storage of soy flour in airtight containers for various uses especially in the fortification of cereal-based products. Fine milling of soybean could be carried out by using a recommended 0.425mm mesh size or by repeat milling using the hammer-mill mesh size available. Cooking demonstrations should be intensified applying more than one soybean treatment method to reduce anti-nutritional factors. More emphasis here should be put on incorporation of soybean into local dishes and diets while testing their acceptability. In the enabling environment, access to inputs, with farmers themselves getting involved in seed multiplication should be enhanced. This should go hand in hand with facilitating access to affordable credit facilities and subsidized mineral fertilisers as well as advocating for government control in the soybean value chain through the farmer input support programme (FISP) or any other future government support to farmers. Meanwhile post-harvest storage of some soybean for

**Table 12. Enabling framework.**

| Food Systems Category | Status | Strategy | Stage |
|---|---|---|---|
| **Socioeconomic factors** | | | |
| Study districts | All districts under study particularly Petauke had reduced chances of soybean processing and utilization | Intensify production, processing and utilization in all the districts<br>In districts like Petauke, wards that lack land and have poor soils could be used as market avenues for locally processed soybean products | Production, distribution, acquisition & consumption |
| Socioeconomic status | Higher social hierarchy exhibited by owning a bed was significantly associated with soybean processing and utilization | Higher social hierarchy exhibited by owning a bed could have been brought about by the engagement in soybean production. This is because those who owned a bed and grow soybean were not significantly different from those who owned a bed and utilized soybean. | Production, Distribution, acquisition & Consumption |
| Ethnic affiliation | Being Chewa was significantly associated with soybean processing and utilization | Promote soybean processing and utilization among all ethnic groups to improve food security, nutrition and health | Production distribution & consumption |
| Awareness | Lack of knowledge about health, nutrition benefits and processing methods | Intensify sensitizations on the benefits of soybean, correct processing methods and application in various household recipes to enhance utilization | Soybean distribution & consumption |
| Gender | Low male in processing and utilization of soybean. Low involvement of female headed households in production and therefore processing and utilization of soybean. | Enhance participation of males in soybean processing and utilization. Enhance participation of female headed households in soybean production processing and utilization. | Soybean distribution & consumption |
| | | Advocate for targeting more social protection programs such as farmer input support program (FISP) with soybean pack to women. | |
| Age | Low participation of the old aged in soybean production, processing and utilization | Make social protection nutrition sensitive to encourage the aged to buy nutritious soy products | Soybean acquisition & consumption |
| Time | Lack of time to prepare soybean products, small household size and labour intensity of methods | Encourage households to adopt low labour intensity low time consuming methods such as heat treating and milling enough soybean in advance for use in household recipes | Soybean distribution & consumption |
| Market | "If we can't sell it, we can't grow it". Lack of or unreliable soybean output markets in some wards | Enhance linkages to soybean output markets. Encourage privately owned soybean processing equipment such as solar harmer mills and extruders in the rural districts<br>Promote incorporation of soybean into local dishes and diets by tracking and trailing possible recipes while testing their acceptability. | Soybean distribution |
| Farm diversification | Off farm income and livestock ownership positively influence soybean production, processing and utilization | Encourage farm diversification such as ownership of livestock to improve incomes among farmers that could be used to boost soybean production and utilization | Soybean production & Consumption |
| Land and soil | Lack of land and poor soils | Encourage selling soybean and soybean products to wards with lack of land and poor soils | Soybean distribution |
| Policy | Lack of government control on soybean marketing | Advocate for recognition of soybean by government in the food systems through the FISP pack or any other future support programmes to small holder farmers | Soybean production |

future use without leaving behind the male folks should be encouraged. Collective action, full scale community involvement and ownership through groups, associations or cooperatives should be exhilarated, while building capacity through training, information exchange, market development as well as creation of new products, development of cottage industries and inter district trade should all be enhanced. In districts like Petauke, wards that lack land and have poor soils should be target markets for cottage industries' products from other districts. In addition, more public-private partnerships should be created along the soybean value chain. Socioeconomic factors ought to be equally enhanced. Firstly there is need to

promote soybean processing and utilization among all ethnic groups to improve coverage in food security, nutrition and health. Participation of males in soybean processing and utilization as well as female headed households in soybean production and therefore processing and utilization should be enhanced. Social protection coverage should be escalated and made nutrition sensitive to encourage the aged and other vulnerable people in the community to buy nutritious soy products. Low labour intensity, low time consuming methods such as heat treating and milling enough soybean in advance for use in household recipes should be encouraged. Solar hammer-mills dotted around the country in rural areas should be encouraged to mill soybean.

In the medium or long term, linkages to soybean output markets should be heightened by advocating for import substitution agreements emphasizing on the replacement of imports such as TSP with domestically produced goods. Local private sector players venturing into the manufacturing of TSP in the rural districts to fill in the supply gap should be supported. TSP manufacturing ventures could be situated in areas connected to the national power grid within the district to efficiently operate. More public-private partnerships in soybean value chain should be created. Farm diversification such as ownership of livestock to improve incomes among farmers ultimately boosting soybean production and utilization should be stimulated. Overall, integrated programs involving all stakeholders in the soybean value chain should be promoted [5].

## Study limitations

The findings of this study could be applied to enhance soybean processing and utilization in similar settings. The study however was conducted in selected districts in the Eastern Province of Zambia. Therefore generalizability should be limited to the districts under study.

## Supporting information

**S1 Fig. Original map of Eastern Province.**
(PNG)

**S1 Table. Data enhancing soybean processing and utilization DTA file.**
(DTA)

## Acknowledgments

We would like to thank the Government of the Republic of Zambia through the Ministry of Agriculture for their direct support to this study through the District Agricultural Coordinators and the Camp officers. Many thanks also go to the Ministry of Health particularly Rural Health Facilities for releasing some staff to be part of the enumerators. Special thanks go to Survey enumerators from five Wards in Petauke namely Chilimanyama, Msumbazi, Nyika, Mateyo Mzeka and Ongolwe. Others are from five wards in Katete specifically; Mkaika, Katiula, Mwandafisi, Chimutende and Matunga. There were also five camp officers from Chitando, Makangila, Msanga, Nthope and Mboza Wards in Chipata. Our appreciation go to the headmen who participated in Focus group discussions (FGDs). These were from Chilimanyama, Musumbazi, Katiula, Chimutende, Nthope and Makangila wards. We should equally give credit to the two Engineers; Schultz Shangala and Funduluka Shangala the assistant data managers and analysts for formulating the structured questionnaire using the KoboCollect v2022.2.3 and later analysing the data. Lastly, we would like to thank the Levy Mwanawasa Medical University for the support in this noble cause.

## Author Contributions

**Conceptualization:** Priscilla Funduluka.

**Data curation:** Priscilla Funduluka.

**Formal analysis:** Priscilla Funduluka.

**Funding acquisition:** Priscilla Funduluka.

**Investigation:** Priscilla Funduluka.

**Methodology:** Priscilla Funduluka.

**Project administration:** Priscilla Funduluka.

**Resources:** Priscilla Funduluka, Natasha Muchemwa Mwila.

**Software:** Priscilla Funduluka.

**Supervision:** Priscilla Funduluka, Natasha Muchemwa Mwila.

**Validation:** Priscilla Funduluka.

**Visualization:** Priscilla Funduluka.

**Writing – original draft:** Priscilla Funduluka.

**Writing – review & editing:** Priscilla Funduluka, Twambo Hachibamba, Mercy Mukuma, Phoebe Bwembya, Regina Keith, Chiza Kumwenda, Natasha Muchemwa Mwila.

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
