## [Decision Letter · Decision Letter 0]

2 May 2023

PONE-D-23-04984Enhancing household Soybean Processing and Utilization in the Eastern Province of Zambia, a concurrent triangulation study designPLOS ONE

Dear Dr. Funduluka,

Thank you for submitting your manuscript to PLOS ONE. After careful consideration, we feel that it has merit but does not fully meet PLOS ONE’s publication criteria as it currently stands. Therefore, we invite you to submit a revised version of the manuscript that addresses the points raised during the review process.

Please pay close attention to all of the comments and concerns raised by the reviewers.

Please submit your revised manuscript by Jun 16 2023 11:59PM. If you will need more time than this to complete your revisions, please reply to this message or contact the journal office at plosone@plos.org. Please include the following items when submitting your revised manuscript:A rebuttal letter that responds to each point raised by the academic editor and reviewer(s). You should upload this letter as a separate file labeled 'Response to Reviewers'.A marked-up copy of your manuscript that highlights changes made to the original version. You should upload this as a separate file labeled 'Revised Manuscript with Track Changes'.An unmarked version of your revised paper without tracked changes. You should upload this as a separate file labeled 'Manuscript'.

We look forward to receiving your revised manuscript.

Kind regards,

Emmanuel Oladeji Alamu

Academic Editor

PLOS ONE

Journal Requirements:

"This work was made possible with the financial and substantive support of the GIZ in Lusaka, Zambia.  The GIZ were also involved in the Conceptualization, Visualization and shaping the methodology."

"I hereby declare, on behalf of all authors, that there are no financial, personal, or professional interests that could be construed to have influenced the work."

5. Please amend your list of authors on the manuscript to ensure that each author is linked to an affiliation. Authors’ affiliations should reflect the institution where the work was done (if authors moved subsequently, you can also list the new affiliation stating “current affiliation:….” as necessary).

Additional Editor Comments:

Dear Authors,

Please pay close attention to all of the comments and concerns raised.

Reviewers' comments:

Reviewer's Responses to Questions

**Comments to the Author**

1. Is the manuscript technically sound, and do the data support the conclusions?

Reviewer #1: Yes

Reviewer #2: No

Reviewer #3: Yes

2. Has the statistical analysis been performed appropriately and rigorously? 

Reviewer #1: No

Reviewer #2: No

Reviewer #3: Yes

3. Have the authors made all data underlying the findings in their manuscript fully available?

Reviewer #1: No

Reviewer #2: No

Reviewer #3: Yes

4. Is the manuscript presented in an intelligible fashion and written in standard English?

Reviewer #1: No

Reviewer #2: No

Reviewer #3: Yes

5. Review Comments to the Author

Reviewer #1: Thank you authors. The manuscript is well written and it is important for improving the growth of the soyabean.

- I would request to go through the English language of the manuscript

- Please add the validity and reliability of data.

- Why verbal consent? Why not written?

- Add limitations of the study in the results and discussion section, including implication

- If possible, please link the study with nutrition

Reviewer #2: The paper entitled enhancing household soybean processing in Eastern Province of Zambia, a concurrent triangulation study design though covers good understanding on subject regarding soybean household processing in Eastern Province of Zambia but it lacks defined objectives of investigative study.

The designed query has several technical flaws. For instance, though soybean is rich in well balanced essential amino acid content but it lacks sulfur containing essential amino acids methionine. It is rich in anti nutrients which is a serious matter of concern as presence of anti-nutrients interferes with the absorption of digested nutrients in the human body. Above all, anti nutrients are not completely removed by common domestic processing method. Prolong soaking of soybean seeds and soybean fermentation are two important methods which are being used to decrease the anti nutrients content of soybean seeds but this requires technical expertise which most of the household individuals lack.

In simple words even if soybean seeds yield increases several folds, the consumption of domestically processed soybean products is not adequately nourishing on nutritive grounds which would be a serious matter of concern for sustaining natural health in the Zambia s soybean domestically processed products consumers population in long run.

As far as soybean oil is concerned, maize which is another major crop of Zambia, produces more healthy oil by using technically simpler method in cost effective manner.

Method of data collection in the submitted work is not scientifically authentic and lacks the provision of identifying and limiting the errors which questions the reproducibility of the investigative work

On the basis of above mentioned grounds, I would recommend major revision of the submitted work while addressing the above raised points.

Reviewer #3: Review: Enhancing household Soybean Processing and Utilization in the Eastern Province of Zambia, a concurrent triangulation study design

1. Title: The title is adequate.

2. Abstract: The abstract is adequately addressed.

3. Introduction: The introduction is adequate.

4. Basic Reporting

The English language structure is professional, clear and unambiguous throughout except in few lines.

The authors were consistent with one style of paragraph.

Page 10: The equation by Yamane (1967) is not sharp, do the needful.

All Table titles should have colon after the number before title. For example; Table 1: ……

Thank you for providing literature and references

5. Experimental design

The research is original primary one and within Aims and Scope of the Journal.

Research question is well defined, relevant & meaningful and it stated how research fills an identified knowledge gap. The experimental design was clearly stated. Rigorous investigations were performed to a high technical & ethical standard.

6. Validity of the Findings

Impact and novelty are assessed. Negative/inconclusive results accepted. Meaningful replication was encouraged since the experimental design was clearly stated; rationale & benefit to literature is clearly stated.

All underlying data have been provided; they are robust, statistically sound, & controlled.

Conclusion: This should address the summary of the objectives outlined for the study, linked to original research question & limited to supporting results. Conclusions are not well stated,

Thus the authors should recast this.

6. PLOS authors have the option to publish the peer review history of their article (what does this mean?). If published, this will include your full peer review and any attached files.

Reviewer #1: No

Reviewer #2: **Yes: **Dr Faiza Abdur Rab

Reviewer #3: **Yes: **Dr. Francis Chigozie Okoyeuzu

<quillbot-extension-portal></quillbot-extension-portal>

---

## [Author Response · Author response to Decision Letter 0]

20 May 2023

Response to Reviewers

Review comment 

The English language structure is professional, clear and unambiguous throughout except in few lines. 

Manuscript has been reviewed to rewrite the few sentences that ambiguous

The equation by Yamane (1967) is not sharp, do the needful. 

The equation by Yamane has been made sharp and clear page 4

All Table titles should have colon after the number before title. For example; Table 1: …… 

All the table titles now have a full stop as specified by the journal after the number before the title pages 6, 8, 9, 11, 13

Conclusions are not well stated. Thus the authors should recast this. 

Conclusion has been recast page 15

It lacks defined objectives of investigative study 

The purpose of the enquiry has been stated on page 3

The designed query has several technical flaws. For instance, though soybean is rich in well balanced essential amino acid content but it lacks sulfur containing essential amino acids methionine. It is rich in anti-nutrients which is a serious matter of concern as presence of anti-nutrients interferes with the absorption of digested nutrients in the human body. Above all, anti-nutrients are not completely removed by common domestic processing method.

If possible, please link the study with nutrition 

These have been addressed on page 3, 12.

Method of data collection in the submitted work is not scientifically authentic and lacks the provision of identifying and limiting the errors which questions the reproducibility of the investigative work

Please add the validity and reliability of data 

Identifying and liming errors have been addressed on pages 4 and 5

I would request to go through the English language of the manuscript 

The English language of the manuscript has been worked on

Why verbal consent? Why not written? 

Obtaining written consent was not practical because the consent form was in-bult in the digital questionnaire created in the Kobocollect app. Verbal consent in this case followed the same requirements as written consent such as providing information, answering questions and respecting the right to withdraw. The verbal consent response was recorded for each participant.

Add limitations of the study in the results and discussion section, including implication 

These have been added on page 15

Competing Interests Recast to:

"I hereby declare, on behalf of all authors, that there are no financial, personal, or professional interests that could be construed to have influenced the work." "This does not alter our adherence to PLOS ONE policies on sharing data and materials.”

Authors contributions Edited on page 16

---

## [Decision Letter · Decision Letter 1]

7 Jun 2023

PONE-D-23-04984R1Enhancing household Soybean Processing and Utilization in the Eastern Province of Zambia, a concurrent triangulation study designPLOS ONE

Dear Dr. FUNDULUKA,

Thank you for submitting your manuscript to PLOS ONE. After careful consideration, we feel that it has merit but does not fully meet PLOS ONE’s publication criteria as it currently stands. Therefore, we invite you to submit a revised version of the manuscript that addresses the points raised during the review process.

We look forward to receiving your revised manuscript.

Kind regards,

Emmanuel Oladeji Alamu

Academic Editor

PLOS ONE

Journal Requirements:

**Additional Editor Comments:**

To improve the quality of the paper, please carefully address the key concerns raised by Reviewer 2, especially the following:

1. information on the evaluation of the effect of various domestic processing methods on the anti-nutrient content of soybeans and the nutritional composition of various domestically produced soybean products. (If these details are unavailable, please note them as a limitation of the study. I will suggest that you include study limitations and recommendations.

2. It would be helpful if the Average Diet Intake Menu Compositions (also in terms of nutritional composition) of an adult (Man and Woman respectively) and a child (Boy and Girl respectively) were included in the paper as a prerequisite for assessing the suitability of the Domestically Prepared Soybean Products, discussed in the paper, for inclusion in the Daily Diet Intake Menu of adults and children while minimizing the risk of Malnutrition in the long term.

3. It is of the utmost importance to include and discuss in the paper the efficacy and dependability of methods recommended for domestic use to remove Anti-Nutrients from Soybean Products mentioned in the paper, after determining their effects on the nutritional composition and digestibility of the Soybean Products to be prepared domestically.

4.It is essential to explain in the paper why Soybean-based Domestically Prepared Products were chosen for study to determine their suitability for regular use over other comparable agricultural products that can be grown cost-effectively and are suitable for domestic processing but lack anti-nutrients.

Reviewers' comments:

Reviewer's Responses to Questions

**Comments to the Author**

1. If the authors have adequately addressed your comments raised in a previous round of review and you feel that this manuscript is now acceptable for publication, you may indicate that here to bypass the “Comments to the Author” section, enter your conflict of interest statement in the “Confidential to Editor” section, and submit your "Accept" recommendation.

Reviewer #1: All comments have been addressed

Reviewer #2: (No Response)

Reviewer #3: All comments have been addressed

2. Is the manuscript technically sound, and do the data support the conclusions?

Reviewer #1: Yes

Reviewer #2: No

Reviewer #3: Yes

3. Has the statistical analysis been performed appropriately and rigorously? 

Reviewer #1: Yes

Reviewer #2: No

Reviewer #3: Yes

4. Have the authors made all data underlying the findings in their manuscript fully available?

Reviewer #1: (No Response)

Reviewer #2: No

Reviewer #3: Yes

5. Is the manuscript presented in an intelligible fashion and written in standard English?

Reviewer #1: Yes

Reviewer #2: No

Reviewer #3: Yes

6. Review Comments to the Author

Reviewer #1: Thank you for responding the suggestions. I would suggest to rigorously review and edit the English language.

Reviewer #2: The revised paper entitled Enhancing household Soybean Processing and Utilization in the Eastern Province of Zambia, a concurrent triangulation study design fails to accommodate all aspects of academic concerns raised in my earlier review.

The manuscript lacks scientifically designed objectives of the study besides indicating inadequate literature review on relevant aspects of the topic while over emphasizing on statistical analysis of inappropriate and insufficient data which can mislead the drawn conclusion.

A very important aspect which is found missing in the investigative study was to evaluate the effect of different domestic processing methods on Soybean s Anti Nutrient Contents and on Nutritional Composition of different Domestically Produced Soybean Products. It would be helpful if Average Diet Intake Menu Compositions ( also in terms of nutritional composition) of an adult ( Man and Woman respectively) and a child ( Boy and Girl respectively ) are mentioned in the paper, a pre requisite to assess the suitability of the Domestically Prepared Soybean Products, discussed in the paper, for adding in Daily Diet Intake Menu of adults and children while avoiding the risk of Mal Nutrition in them in long run. There is an utmost need to add and to discuss in the paper the efficiency and reliability of methods being recommended for domestic use to remove Anti-Nutrients from Soybean Products referred in the paper after assaying their effects on nutritional composition and digestibility of the Soybean Products to be prepared domestically.

It is important to discuss in the paper why have Soybean based Domestically Prepared Products been preferred for study to assess their suitability for regular use over other comparable agricultural products which can be grown cost effectively and are suitable for domestic processing but lacks Anti Nutrients.

On the basis of my above mentioned findings, I would recommend major revision of manuscript while addressing the points mentioned in this review of the Submitted Revised Manuscript.

Reviewer #3: The authors have done all the requested and necessary corrections, thus the paper can be accepted for publication

7. PLOS authors have the option to publish the peer review history of their article (what does this mean?). If published, this will include your full peer review and any attached files.

Reviewer #1: No

Reviewer #2: **Yes: **Dr Faiza Abdur Rab

Reviewer #3: **Yes: **Francis Chigozie Okoyeuzu

While revising your submission, please upload your figure files to the Preflight Analysis and Conversion Engine (PACE) digital diagnostic tool, https://pacev2.apexcovantage.com/. PACE helps ensure that figures meet PLOS requirements. To use PACE, you must first register as a user. Registration is free. Then, login and navigate to the UPLOAD tab, where you will find detailed instructions on how to use the tool. If you encounter any issues or have any questions when using PACE, please email PLOS at figures@plos.org. Please note that Supporting Information files do not need this step.<quillbot-extension-portal></quillbot-extension-portal>

---

## [Author Response · Author response to Decision Letter 1]

22 Jul 2023

Reference list has been reviewed.

1. Information on the evaluation of the effect of various domestic processing methods on the anti-nutrient content of soybeans and the nutritional composition of various domestically produced soybean products. (If these details are unavailable, please note them as a limitation of the study.

More literature has been added highlighted in blue and grey in the introduction

2. I will suggest that you include study limitations and recommendations. 

Added as part of introduction highlighted in purple

These have been included and are appearing after conclusion highlighted in yellow 

2. It would be helpful if the Average Diet Intake Menu Compositions (also in terms of nutritional composition) of an adult (Man and Woman respectively) and a child (Boy and Girl respectively) were included in the paper as a prerequisite for assessing the suitability of the Domestically Prepared Soybean Products, discussed in the paper, for inclusion in the Daily Diet Intake Menu of adults and children while minimizing the risk of Malnutrition in the long term. 

These have been added as part of the introduction highlighted in light blue

3. It is of the utmost importance to include and discuss in the paper the efficacy and dependability of methods recommended for domestic use to remove Anti-Nutrients from Soybean Products mentioned in the paper, after determining their effects on the nutritional composition and digestibility of the Soybean Products to be prepared domestically.

A very important aspect which is found missing in the investigative study was to evaluate the effect of different domestic processing methods on Soybean s Anti Nutrient Contents and on Nutritional Composition of different Domestically Produced Soybean Products.

There is an utmost need to add and to discuss in the paper the efficiency and reliability of methods being recommended for domestic use to remove Anti-Nutrients from Soybean Products referred in the paper after assaying their effects on nutritional composition and digestibility of the Soybean Products to be prepared domestically.

These have added to the introduction highlighted in grey

It is important to discuss in the paper why have Soybean based Domestically Prepared Products been preferred for study to assess their suitability for regular use over other comparable agricultural products which can be grown cost effectively and are suitable for domestic processing but lacks Anti Nutrients. 

Added to the introduction highlighted in dark and light blue 

4.It is essential to explain in the paper why Soybean-based Domestically Prepared Products were chosen for study to determine their suitability for regular use over other comparable agricultural products that can be grown cost-effectively and are suitable for domestic processing but lack anti-nutrients. 

Reviewer #1: Thank you for responding the suggestions. I would suggest to rigorously review and edit the English language. 

The English language has been reviewed

Reviewer #2: The revised paper entitled Enhancing household Soybean Processing and Utilization in the Eastern Province of Zambia, a concurrent triangulation study design fails to accommodate all aspects of academic concerns raised in my earlier review.

The manuscript lacks scientifically designed objectives of the study besides indicating inadequate literature review on relevant aspects of the topic while over emphasizing on statistical analysis of inappropriate and insufficient data which can mislead the drawn conclusion. 

Objectives are highlighted in red. They appear just before methodology

---

## [Editor Report · Decision Letter 2]

25 Jul 2023

PONE-D-23-04984R2Enhancing household Soybean Processing and Utilization in the Eastern Province of Zambia, a concurrent triangulation study designPLOS ONE

Dear Dr. FUNDULUKA,

Thank you for submitting your manuscript to PLOS ONE. After careful consideration, we feel that it has merit but does not fully meet PLOS ONE’s publication criteria as it currently stands. Therefore, we invite you to submit a revised version of the manuscript that addresses the points raised during the review process.

Please rewrite the study's objective in full sentences rather than as a subheading. This is not a dissertation. 

In addition, because you have not yet responded to the comment on editing, the paper requires extensive language editing. 

Please submit your revised manuscript by Sep 08 2023 11:59PM. If you will need more time than this to complete your revisions, please reply to this message or contact the journal office at plosone@plos.org. Please include the following items when submitting your revised manuscript:A rebuttal letter that responds to each point raised by the academic editor and reviewer(s). You should upload this letter as a separate file labeled 'Response to Reviewers'.A marked-up copy of your manuscript that highlights changes made to the original version. You should upload this as a separate file labeled 'Revised Manuscript with Track Changes'.An unmarked version of your revised paper without tracked changes. You should upload this as a separate file labeled 'Manuscript'.If applicable, we recommend that you deposit your laboratory protocols in protocols.io to enhance the reproducibility of your results. Protocols.io assigns your protocol its own identifier (DOI) so that it can be cited independently in the future. For instructions see: https://journals.plos.org/plosone/s/submission-guidelines#loc-laboratory-protocols. Additionally, PLOS ONE offers an option for publishing peer-reviewed Lab Protocol articles, which describe protocols hosted on protocols.io. Read more information on sharing protocols at https://plos.org/protocols?utm_medium=editorial-email&utm_source=authorletters&utm_campaign=protocols.

We look forward to receiving your revised manuscript.

Kind regards,

Emmanuel Oladeji Alamu

Academic Editor

PLOS ONE

Journal Requirements:

While revising your submission, please upload your figure files to the Preflight Analysis and Conversion Engine (PACE) digital diagnostic tool, https://pacev2.apexcovantage.com/. PACE helps ensure that figures meet PLOS requirements. To use PACE, you must first register as a user. Registration is free. Then, login and navigate to the UPLOAD tab, where you will find detailed instructions on how to use the tool. If you encounter any issues or have any questions when using PACE, please email PLOS at figures@plos.org. Please note that Supporting Information files do not need this step.<quillbot-extension-portal></quillbot-extension-portal>

---

## [Author Response · Author response to Decision Letter 2]

8 Aug 2023

Please rewrite the study's objective in full sentences rather than as a subheading. This is not a dissertation. 

Action

This has been done. It is highlighted in yellow at the end of the introduction, just before methodology

In addition, because you have not yet responded to the comment on editing, the paper requires extensive language editing.

Action

Extensive language editing has been done throughout the paper.

---

## [Editor Report · Decision Letter 3]

9 Aug 2023

Enhancing household Soybean Processing and Utilization in the Eastern Province of Zambia, a concurrent triangulation study design

PONE-D-23-04984R3

Dear Ms. FUNDULUKA,

We’re pleased to inform you that your manuscript has been judged scientifically suitable for publication and will be formally accepted for publication once it meets all outstanding technical requirements.

Kind regards,

Emmanuel Oladeji Alamu

Academic Editor

PLOS ONE
---

## [Editor Report · Acceptance letter]

15 Sep 2023

PONE-D-23-04984R3 

Enhancing household Soybean Processing and Utilization in the Eastern Province of Zambia, a concurrent triangulation study design 

Dear Dr. Funduluka:

I'm pleased to inform you that your manuscript has been deemed suitable for publication in PLOS ONE. Congratulations! Your manuscript is now with our production department. 

Kind regards, 

on behalf of

Dr. Emmanuel Oladeji Alamu 

Academic Editor

PLOS ONE